# Patient Preferences for Long-Term Implant Care in Cochlear, Glaucoma and Cardiovascular Diseases

**DOI:** 10.3390/ijerph20146358

**Published:** 2023-07-13

**Authors:** Sabine Schulz, Laura Harzheim, Constanze Hübner, Mariya Lorke, Saskia Jünger, Annika Buchholz, Stefanie Frech, Melanie Steffens, Christiane Woopen

**Affiliations:** 1Cologne Center for Ethics, Rights, Economics, and Social Sciences of Health (CERES), University of Cologne and University Hospital of Cologne, Universitätsstraße 91, 50931 Cologne, Germany; laura.harzheim@uni-koeln.de; 2Center for Life Ethics, University of Bonn, 53113 Bonn, Germany; constanze.huebner@uni-bonn.de (C.H.); chwoopen@uni-bonn.de (C.W.); 3Faculty of Engineering and Mathematics, University of Applied Sciences and Arts (HSBI), 33619 Bielefeld, Germany; mariya.lorke@hsbi.de; 4Department of Community Health, University of Applied Health Sciences Bochum, Gesundheitscampus 6-8, 44801 Bochum, Germany; saskia.juenger@hs-gesundheit.de; 5Department of Otolaryngology, Hannover Medical School, Carl-Neuberg-Str. 1, 30625 Hannover, Germany; buchholz.annika@mh-hannover.de (A.B.); steffens.melanie@mh-hannover.de (M.S.); 6Department of Ophthalmology, Rostock University Medical Center, Doberaner Str. 140, 18057 Rostock, Germany; frechstefanie@web.de

**Keywords:** sustainability, compatibility, implant failure, durability, decision making, information needs, technological progress, treatment success, reimplantation

## Abstract

In the process of developing and implementing innovative implant technologies the consideration of patient preferences can be beneficial for patients, doctors and developers. Nevertheless, in existing literature, there is still scarce knowledge of patients’ perspectives on long-term implant care. In this study, three discrete choice experiments (DCEs) were conducted in the context of cochlear implants (CI, *n* = 92), glaucoma implants (GI, *n* = 21) and cardiovascular implants (CVI, *n* = 23), examining the relative importance of attributes of long-term implant care from the patients’ perspective. The participants chose between differently shaped options for implant-related care. The attributes of these care options were generated and selected based on previous literature reviews, group discussions and a diary study with patients. The choice data were analyzed via binary logit regression. In CI-DCE, the technological compatibility of the implant with newer implant models, accessories or devices from other manufacturers was highly valued by participants, whereas in GI-DCE the (in)dependency on glaucoma medication post-implantation had the greatest influence on participants’ choice behavior. In CVI-DCE, the attribute with the highest relative importance related to the means of securing long-term treatment success. In all three DCE, shared decision making was relatively important for participants. Our results emphasized the importance of an adequate transfer of technological advancements in implant care for promoting patient benefits, such as the availability of comprehensible, understandable, high-quality information about current developments. Similarly, promoting technological health literacy and further pushing the technological compatibility, durability and safety of implants are directions for future implant development in accordance with patients’ preferences. Therefore, the participation of implant wearers in the development process is encouraged.

## 1. Introduction

Implant-based interventions offer various possibilities for the treatment of chronic diseases or physiological conditions. They involve the insertion of technological devices into the human body for long-lasting (or permanent) retention [1]. Advancements in implant technology continuously expand the scope of applications, e.g., decreasing the possible age at first implantation [2]. Alongside a generally increasing life expectancy [3], this can bring about a prolonged implant wearing time, thus elevating the requirements for the average service life of an implant, entailing the long-term involvement of implant wearers in healthcare services and implant management. Therefore, implant-based interventions should also be considered from a long-term perspective. Based on three exemplary fields of implant care for chronic diseases or physiological conditions, this study aims to provide corner points for generating knowledge on long-term patient perspectives and preferences. The following three implant types are referred to in this study: (1) cochlear implants (CIs), which enable persons with hearing loss to (re)gain their sense of hearing; (2) glaucoma implants (GI) that reduce intraocular pressure and, therewith, play an important part in glaucoma management; and (3) passive cardiovascular implants (CVI), such as coronary stents or valve implants for treating cardiovascular diseases (The study presented in this work was part of a Germany-wide collaborative research project (RESPONSE) examining how innovative implant technologies can be successfully and sustainably developed and implemented in the healthcare practice. In this study, the RESPONSE project focused on innovative implant technologies for the treatment of cardiovascular diseases, cataracts and glaucoma, as well as hearing loss and deafness, as distinctive medical application fields characterized by high epidemiological prevalence combined with a strong technology-driven need for innovation, as well as high market volumes and above-average growth rates from a global perspective [4]. Due to the chronic character of the underlying diseases or health conditions, these fields are also particularly well suited to the investigation of implant care from a long-term perspective.). In Germany alone, these may affect 800,000 patients annually who suffer from vascular diseases, 440,000 patients with diseases of the eyes and, potentially, about 14 million people affected by hearing disorders [5]. The focus of implant treatment is on maintaining the highest possible quality of life for patients, well into old age [5]. Especially given the scarce resources in the healthcare system necessitating a prioritization and rationing of healthcare services and goods, it is advisable to only launch or offer worthwhile products or services that have compelling benefits for patients [6,7]. In the existing literature, there is still scarce knowledge of the long-term experiences of individuals wearing CI, GI and CVI [8], and of how such experiences may affect patient preferences. Such knowledge may be crucial for patient–doctor communication and implant development, since decision making on implant treatment is solely one initial step in implant care. In a field characterized by fast-paced technological innovations, incorporating patient preferences into the development and/or implementation of healthcare services or products can facilitate the realization of patient benefits and acceptance and, overall, improves patient-centered and demand-led care.

In order to generate knowledge on patient preferences in long-term implant care, discrete choice experiments (DCEs) among individuals wearing CI, GI and CVI were conducted and analyzed for the purposes of this study. DCEs are a tool widely used in health economics to assess patient preferences [9]. In the present context, they represent a proven method to weigh patient-sided outcomes and allow the demonstration of additional patient benefits from services or innovative technologies [6]. Aspects that contribute to long-term benefits, reliability and sustainability are of great interest when designing long-term implant care, since implants affect an individual’s everyday life and body perceptions, as well as the way societies define health, disease and disability [8]. Previous research has shown that the assessment of an implant from the implant wearer’s perspective often exceeds its mere clinical characteristics, including the individual’s experiences of implant management in everyday life or concomitant healthcare services [8,10]. Due to the increasing technological development in the field of implants over the past two decades, the number of studies in the implant fields investigated here is limited. Studies published so far have focused mainly on clinical parameters, while individual patient perspectives and participatory research designs have played a subordinate role. Considering this, the aim of this study was to address this research gap and examine the relative importance of different aspects of long-term implant care from the perspective of patients’ preferences. The goal was to provide patient-centered insights into how innovative implant technologies can be successfully and sustainably developed and implemented in healthcare practice. The aspects addressed in this study were elaborated in a previous, extensive qualitative study (group discussions and diary study with 39 implant wearers) [10] and transferred into the DCE methodology. This approach enabled the definition of attributes and the design of the DCEs in a research context lacking relevant existing literature. The results from these DCEs can therefore provide novel, systematic information about patient preferences based on their long-term experiences in the exemplary clinical fields and contribute to patient orientation, preference-sensitive education and shared decision making in implant care. Furthermore, the implications for optimizing implant development and long-term implant care, as well as impulses for prioritizing healthcare services or goods while considering patient-sided benefits, could be provided.

## 2. Materials and Methods

In order to systematically assess patient preferences regarding factors of long-term implant care, three DCEs implemented in an online survey format (one in each clinical field) were conducted. The design of the DCEs was developed based on a synthesis of existing research in the literature and data from previous qualitative research. After the analysis of the literature and qualitative data (collected in group discussions and a diary study with implant wearers), research gaps were identified and research questions were formulated.

### 2.1. Discrete Choice Experiments

Participants in DCEs are repeatedly presented with choice situations, containing hypothetical scenarios for the participants to choose from. In this study, such scenarios represent differently shaped options for implant-related care, denoted as care scenarios (CSs). CSs comprise the same set of *attributes*, but differ in the attributes’ specification, so-called *levels* (e.g., attribute: technological compatibility—level 1: not compatible, level 2: compatible). The composition of the levels in each CS is systematically varied, according to an experimental design plan.

Following best practice guidance [11,12,13] and fundamental methodological literature [14,15,16,17,18], a DCE was developed for each clinical field.

#### 2.1.1. Attributes and Levels

After determining the research objectives, the attributes and levels were developed. Based on a previously conducted systematic literature review [8] and qualitative studies with implant wearers [10], a list of relevant attributes and levels was derived. Parameters related to the definition of the attributes were either discussed in the group discussions with implant wearers, referred to in the diary study/existing literature, or both. The list of attributes contained both attributes based on retrospective experiences, as well as attributes based on preferences concerning future developments in technology and care. Out of these, a final compilation of attributes and levels was selected according to their applicability with regard to the DCE format. Due to the different amount of data and the specifics of each implant type, the procedure for attribute selection varied depending on the field of application. For instance, the life span of the implant and the associated treatment pathway (e.g., lifelong medication or reimplantation) were important especially in the CVI field, while in the CI field, the treatment team and aftercare were relevant due to the technical circumstances and the involvement of different treatment groups (e.g., physician, hearing aid technician, manufacturer, etc.). Since the focus of this study was on long-term implant care, the final list contained attributes that addressed long-lasting impacts on the patient’s everyday life with the implant and were associated with the patient’s ongoing involvement in healthcare and implant management. The list of potential attributes and their levels were extensively discussed with experts in the three implant fields and a final selection was performed. In the end, six attributes in each implant area were selected. This number is in line with the majority of healthcare-related DCEs [19].

To ensure the plausibility, comprehensibility and relevance of the attributes and levels, five cognitive interviews were conducted (approx. 60 min, conducted in person with adult participants, convenience sample, no implant wearers or patients). In addition, experts in the respective clinical fields were consulted to validate the selection of the attributes and levels (experts: Prof. Tobias Schilling, Katharina Band, M.Sc. (Clinic for Heart, Thorax and Vascular Surgery, Hannover Medical School), Anne Wolf (Department for Cardiology, Center of Internal Medicine, Rostock University Medical Center), Dr. Stefanie Frech (Department of Ophthalmology, Rostock University Medical Center), Annika Buchholz, Dr. Melanie Steffens (Department of Otolaryngology, Hannover Medical School)). According to the information gained in the pre-testing, the attributes and levels were refined and adapted. An overview of the final attributes and levels in each clinical field is displayed in Table 1.

#### 2.1.2. Experimental Design

The varying attribute levels resulted in 144 hypothetical CSs and, opting for paired choices, 10.296 potential paired choice situations (CI- and GI-DCE), respectively, and 98 hypothetical CSs leading to 4753 potential choice situations (CVI-DCE). Considering the vast amount of hypothetical choice situations, a fractional factorial experimental design was applied to sufficiently estimate all relevant effects, while minimizing the cognitive burden for participants. According to Rose and Bliemer, the “optimal orthogonal design is the most efficient when no parameter priors are assumed” [18] (p. 608). Hence, an optimal orthogonal design plan (OOD) was chosen to maximize information from each choice situation, since in this case no parameter priors were available. Moreover, the focus was on the main effects, which generally make up 70% to 90% of the explained variance [17]. Using the software Ngene [20], the DCEs were constructed according to an OOD plan and a sample of 36 (CI-DCE, GI-DCE), respectively, and 24 (CVI-DCE) choice situations were found to be sufficient for estimating the main effects. With regards to participants’ efficiency and cognitive burden, a blocked format was used, randomly dividing the choice situations into two blocks of 18 (CI-DCE, GI-DCE), respectively, and 12 (CVI-DCE) choice situations each.

With regards to the required sample size, following a rule of thumb approach proposed by Johnson and Orme [21,22], estimating reliable models for the chosen experimental designs requires 84 participants (41.6 participants per block) in CI- and GI-DCE and 125 participants (62.5 participants per block) in CVI-DCE. Other research suggests that precise parameter estimates can be provided with 20–30 respondents per version [16,23].

### 2.2. Data Collection

The DCEs were administered in an online setting via the survey platform Qualtrics [24]. The surveys were available online from November 2021 to March 2022. A German and English version of the CI-DCE survey, exemplary for all three clinical fields, is displayed in the Appendix A. Ethics approval was obtained in November 2020 (Nr: 20-1176_1) by the Medical Faculty of the University of Cologne.

The target groups investigated included patients with a cochlear implant, glaucoma implant or glaucoma disease, and with a passive cardiovascular implant. The study information and call for participation were distributed via cooperating clinical departments, widely established patient forums, initiatives or umbrella organizations for self-help groups. With regards to GI, the recruitment strategy was developed and implemented in cooperation with the Department of Ophthalmology, Rostock University Medical Center and the eye clinic of the University Hospital of Cologne. Participants in the cochlear-DCE were recruited in cooperation with the Department of Otolaryngology, Hannover Medical School. Along with the study information, an anonymous link and QR code were provided for participants to access the online survey.

Before starting the DCEs, participants received information on the study objective, procedure, data processing and data protection. After obtaining consent and assessing the participants’ eligibility (see the criteria in Table 2), they were randomly assigned to one block of the DCE and answered the choice tasks presented in a random order. Finally, the participants were asked about their age and gender. No other personal or condition-specific data was requested, with regard for participants’ anonymity.

### 2.3. Analysis

The theoretical foundation for analyzing choice data is provided by random utility theory (RUT), according to which choice behavior is determined by those alternatives which provide the greatest possible benefits for the decision maker [6,16,25,26]. The individual utility of an alternative cannot be observed directly and is understood as a latent construct. Following RUT, latent utilities are composed of a systematic (explainable) component and a random (non-explainable) component. The random component represents unmeasured variation in preferences, which may be due to unobserved additional attributes affecting the choice, measurement error or inter-individual differences in utility (heterogeneity in tastes). Participants’ choice behavior maps the individual utility of an alternative, thereby allowing for the indirect measurement of a significant proportion of the individual utility.

#### 2.3.1. Model of Utility

The latent utility (U_i_) associated with hypothetical care option i was estimated as U_i_ = V_i_ + ε_i,_ where ε_i_ is the random component and V_i_ is the systematic component, assumed to be a composition of all the attribute influences. All the attributes were, a priori, presumed to have a significant impact on the utility. For e.g., in CI-DCE, the utility of hypothetical care option i was estimated as follows (see Equation (1)):U_i_ = V_i_ + ε_i_ = β_1_*decisionSHARED + β_2_*decisionPAT + β_3_*infoMANU + β_4_*infoCLINIC + β_5_* futureoptionPRESERVE + β_6_* technCOMPATIBLE + β_7_* aftercareVARYING + β_8_* educationCOMPR + ε_i_(1)

Detailed models of utility for the GI- and CVI-DCE are presented in Appendix B.

#### 2.3.2. Statistical Analysis

Relating choices to the characteristics of the alternatives available to decision makers, all the while accounting for the clustered data structure, a binomial logit model with robust variance estimation was applied to analyze the choice data, considering the participant’s ID as a repeated measure variable. The statistical analysis was conducted using the software SPSS (IBM SPSS Statistics 28.0.0.0). Beta coefficients indicate the relative contribution of an attribute level regarding the perceived utility of an alternative and the relative impact of an attribute level on choice behavior, respectively.

## 3. Results

In the following, the results of the DCEs for each clinical field are presented separately. It should be noted, that despite extensive recruitment efforts, the required sample sizes could not be reached for the GI- and CVI-DCE. Hence, the analyses in these fields are to be interpreted with caution and the respective results should be considered to be explorative rather than conclusive. Since there is not much research regarding patient preferences with respect to GI or CVI, we decided to include these studies in the current article in the hope of providing the first insights into this research field and providing impetus for further research.

### 3.1. Patient Preferences for Sustainable Implant Care in the Context of Cochlear Implants

Data from 92 individuals (43 female, 49 male), corresponding to a total of 3312 observations were included in the analysis. The mean age was 57 years old (standard deviation (SD): 18.3, range: 19–95). Except for the care relationship in aftercare (*p* = 0.137), all the attributes contributed significantly to the classification of choice behavior (*p* < 0.001). *Age* and *sex* did not yield a significant effect: sex (*p* = 0.334), age (*p* = 0.880). All the model effects are displayed in Section A.1. The odds ratio (OR) for technological compatibility (ref: not compatible) was 4.8, meaning that the likelihood of a CS being chosen was approximately 5 times higher if the CI was compatible with newer implant models and accessories or devices from other manufacturers instead of a non-compatible CI. Considering the model coefficients and odds, technological compatibility had the greatest relative impact on choice behavior, followed by shared decision making, the preservation of accessing alternative treatment options in the future, comprehensive education regarding CI-related adjustments, decisions and innovations, patient-sided decision making and the automatic provision of information by the clinic or manufacturer. All the model coefficients and odds for the cochlear-DCE are shown in Table 3.

### 3.2. Patient Preferences for Sustainable Implant Care in the Context of Glaucoma Implants

Data from 21 individuals (9 female, 12 male), equivalent to a total of 756 observations, were included in the analysis. The mean age of these persons was 57 years old (SD: 13.9, range: 22–78). All the attributes except for the information exchange between health professionals (*p* = 0.814) contributed significantly to the classification of choice behavior (*p* < 0.05). *Age* and *sex* did not yield a significant effect: sex (*p* = 0.613), age (*p* = 0.241). All the model effects are displayed in Section A.2. The highest relative impact on choice behavior was associated with a 75% probability of treatment success (ref: 50% chances) (OR = 5.551), demonstrating that the likelihood of a CS being chosen was approximately 5.5 times higher if the chances of successful treatment (success in the sense of not needing glaucoma medication after 2 years post-implantation) were 75% compared to 50%. Furthermore, the participants preferred statistics and empirical values on implantation in a treating clinic to be available and information going beyond implantation and aftercare (e.g., on nutrition, drops for dry eyes, glasses, etc.) to be provided in the context of implant care from the medical side. This was followed by corrective intervention in the case of implant failure and shared decision making. All the model coefficients and odds for the GI-DCE are shown in Table 4.

### 3.3. Patient Preferences for Sustainable Implant Care in the Context of Cardiovascular Implants

Data from 23 individuals (9 female, 14 male) and, accordingly, a total of 552 observations were included in the analysis. The mean age of these persons was 58 years old (SD: 11.8, range: 29–80). Concerning the model effects, three out of six attributes significantly contributed to the classification of choice behavior, namely decision making, data transparency and the means of maintaining treatment success (*p* < 0.05). The information source (*p* = 0.716), information exchange between health professionals (*p* = 0.818) and invasiveness of the intervention (*p* = 0.113) were not significant, neither were sex (*p* = 0.560) nor age (*p* = 0.170). All the model effects are displayed in Section A.3. With an OR of 11.9 (inverted), the likelihood of a CS being chosen was approximately 12 times higher if treatment success could be secured by means of lifelong medication instead of reimplantation every 10 years. Ensuring treatment success by means of lifelong medication had the highest relative impact on choice behavior, followed by shared and patient-sided decision making and data transparency. All the model coefficients and odds for the cardiovascular-DCE are shown in Table 5.

## 4. Discussion

Our findings provide valuable insights into long-term demand-led implant care from the perspective of everyday experiences of implant wearers. Embedded in existing research, the results from the DCEs in all three categories of implant wearers (CI, GI and CVI) allowed for the formulation of several conclusions valid across the three exemplary clinical fields. These refer to the attributes addressing the access and exchange of information, the patient’s involvement in decision making and the success/risk of the treatment.

The navigation of fast-paced technological and scientific advancements alongside necessary long-term follow-up care and implant management may pose major challenges for implant wearers. In this context, sufficient implant-related information and knowledge is essential for the successful management of the health condition [10], not only with respect to the initial implantation, but also regarding long-term implant care amidst fast-paced technological advancements (often accompanied by a lack of available long-term data, as in the case of GI and CVI [27,28,29]). Our results can offer novel insights on how to adequately address patient information needs. Participants in the DCEs preferred to obtain (comprehensive) information regarding implant management from medical professionals in the context of implant care. This preference may correspond with a wish for reliable and correct information amid a complex and constantly changing information landscape. In several qualitative studies, implant wearers (or the parents of pediatric patients) reported feeling poorly informed, wishing for more in-depth information [30,31,32,33]. This is especially relevant with regards to patients’ autonomy and informed consent when faced with making a decision with long-term consequences [8] in a field with fast-paced technological innovations. Against this background, promoting technological health literacy on an individual, as well as systemic, level could support the appropriate handling of this demanding information landscape, not only with respect to patients’ implant management but also on the way medical professionals provide and convey information [10]. This speaks to the responsiveness of the healthcare system to individual (informational) needs. Implant wearers might be savvy with terminology related to their conditions while having difficulties in other areas, e.g., the appraisal of statistical information or risk communication [10]. Therefore, it can be of great value to take into account and respond to individual needs and preferences in the counseling process, also in cooperation with caregivers and specialists who may be involved in the counseling process [34,35,36]. In a field dominated by ambiguous information and fast-changing evidence, healthcare providers should create efficient frameworks for orientation. Here, the sensitization of the technological layer of information, as well as the provision of advice and training on how to cope with the burden and stress elicited by too much, too complex or too ambiguous information, could promote patient (technological) health literacy and empowerment [10].

Furthermore, the long-lasting introduction of an implant inside the body, its influence on the physical condition and its potential transformational impact on the implant wearer’s everyday life [37,38,39], elucidate the long-term consequential nature of opting for implantation. Our findings demonstrated that patient involvement in decision making, as well as the availability of statistical data regarding the clinics’ experience with the implantation procedure, are highly valued by patients. In order to enable shared decision making, doctors need to be trained in assisting patients in the process of weighing arguments for or against an implant, also thematizing the implications and consequences of a certain decision for long-term implant management and care. In terms of patient-centered and demand-led care, healthcare professionals need to be sensitized to the relevance of individual factors (like living conditions, life expectancy, age and life planning) on patient preferences to adequately address these in the medical consultation, as well as align long-term care and implant management accordingly.

Since implant interventions are shaped by rapid technological developments paired with the irreversibility of the implantation, another important aspect to consider in terms of decision making is (anticipated) decisional regret. Prospective implant wearers have to make a decision in the here and now, anticipating that their form of treatment or implant model may no longer be ‘best practice’ in a few years’ time. This is pronounced by the increasing in vivo lifetime of implants. Examining decisional regret for each implant-specific context (as it is, for example, conducted with respect to surgical decision making [40]) could shed light on how to best support patients in their implant-related decision making and, thereby, strengthen patient autonomy. Therefore, addressing (anticipated) decisional regret in relation to implant interventions is an important subject for future research on how to improve patient-centered implant care.

Apart from the statements related to all three implant types, the analysis of the DCEs also revealed the importance of implant-related aspects, which are specific for each implant field.

Participants in the CI-DCE valued the implant’s compatibility with newer CI models and accessories or devices from other manufacturers the most. A high degree of compatibility would allow CI wearers to benefit from a wider range of technological features without the limitation that each technological equipment should stem from the same manufacturer. It is conceivable how device compatibility could play a significant role on one’s quality of life and social participation in the long-term (e.g., if the implant is compatible with services for CI wearers provided by the entertainment industry or with equipment, such as special microphones facilitating hearing in noisy environments, such as work or school). Such practical implications of an electronic implant’s technological compatibility for implant wearers’ everyday lives are highly instructive. The research on lived experiences or needs and preferences in terms of technological compatibility in everyday implant use (e.g., connectivity of different implant models or accessories, such as headphones) is highly underrepresented in the existing research. The body of literature is focused mainly on the concept of compatibility in the context of (in)compatible medical procedures (e.g., MRI) [41,42,43,44,45] or the biocompatibility of prosthetic devices [46], neglecting the patient’s individual technical preferences, needs and experiences. Furthermore, compatibility also relating to updatability (of the software) is important to provide a similar range of functions to all implant wearers irrespective of the specific implant make (without reimplantation). Depending on the specific (electronic) implant model, the available range of functions and, therefore, the implant wearer’s scope of action and agility in everyday life, might vary. In this regard, issues of equitable access and fairness of distribution arise. In particular, differences in health insurance coverage or hesitant attitudes are highly problematic and require the patient to adopt an assertive and active role in claiming services [10], leaving the potential for discrimination against patients who are less assertive or uncertain about their rights. On another note, CI wearers expressed a clear preference for preserving access to alternative treatment options after implantation. Since inner ear surgery and electrode insertion can cause cochlear damage, affecting residual hearing [47,48], the decision for the introduction of a CI can preclude implant wearers from innovative treatment options based on residual hearing, e.g., combined electric–acoustic stimulation for better speech perception (EAS) [49]. Such irreversibility would increase the dependence of implant wearers on the implant manufacturers’ technological state, which further stresses the relevance of ongoing research and implant technology development efforts in hearing preservation during or following cochlear implantation [47,50,51].

In the case of GI, the prospect of increased treatment success in terms of not needing daily glaucoma medication was strongly valued by participants. This is supported by research stating the certainty of successful outcomes and proven longevity of the treatment effects to be primary motivators for treatment decisions [52]. The daily eye drop regimen in glaucoma management can be perceived as a burden and relates to adherence challenges [53,54], necessitating individual strategies for managing and adhering to daily glaucoma medication [55]. Frustrations with glaucoma medication can reinforce the decision to undergo surgery [54] and potentially underlie patients’ preferences regarding the prospect of long-term liberation from glaucoma medication. However, currently only half of glaucoma drainage devices remain functional after 5 years, highlighting the importance of ongoing research into refining the biomaterials, techniques or shapes of the devices [56]. In the case of implant failure, the participants preferred to restore implant functionality by means of intervention. This may be an indication that patients are open to the possibility of a corrective intervention and could provide impetus for the future development of implants or procedures, which consider this option. The possibility of implant removal, however, did not significantly impact choice behaviour compared to the prospect of the malfunctioning implant remaining in the eye, which might be due to the implant not being physically noticeable [10].

In the case of CVI, ensuring treatment success by means of lifelong medication had the highest relative impact on choice behavior, followed by shared and patient-sided decision making, as well as transparency of data on implantation success. Our findings align more with the characteristics of a surgically implanted mechanical valve prosthesis rather than a bioprosthesis implanted via minimally invasive TAVI [57]: surgically implanted mechanical prosthesis are highly durable but necessitate lifelong anticoagulation medication. To obviate the need for a second valve procedure, mechanical valves are often implanted in younger patients [58]. The implantation of bioprostheses via the TAVI procedure is less invasive compared to chest opening in SAVR and does not demand lifelong anticoagulation medication, but is associated with an increased risk of reintervention over the short term [57]. Hence, from a very simplified viewpoint, the stated preference for lifelong medication compared to reimplantation as means of securing treatment success, alongside the non-significance of the invasiveness of interventions, favors surgically implanted mechanical valves. However, in another preference study, patients put greater value on attributes favoring TAVI, such as a lower mortality rate, reduced procedural invasiveness and quicker time to return to normal life [59]. It has to be noted that these are short-term attributes surrounding the implantation and subsequent recovery. Such differences in preferences related to short-term versus long-term aspects of implant care raise questions as to whether patients lack awareness of the long-term consequences of the intervention or are decidedly focused on the short-term consequences when assessing treatment options. Further technical innovations, such as novel coatings for e.g., which may offer long-lived conditions and low maintenance [60,61] might impact patients’ preferences and decision making in the future, as well as blur the border between short- and long-term perspectives and need to be examined in more detail. The participants in this study had a mean age of 58 and represented a comparatively young group of cardiovascular patients. This may also be a reason that long-term life planning and life expectancy after implantation were given a higher priority and affected the preferences regarding long-term implant care. Furthermore, a preference for information on the immediate medical risks of upcoming procedures provided in medical consultations is conceivable (e.g., with fewer surgery risks and decreased recovery time associated with TAVI), potentially diverting from the relevant long-term aspects.

### 4.1. Study Implications for Future Developments in Long-Term Implant Care

The analysis of all three DCEs revealed some aspects that may provide several starting points for the future of sustainable long-term implant development and care in general.

In view of the rapid technological development, it is important to ensure that implant development takes into account implant wearers’ lifeworlds and that technological progress is as much as possible transferred to their reality of care. Therefore, we highly encourage the participatory involvement of (prospective) implant wearers and their significant others in the development process and implant-related research. Moreover, the importance of data transparency and the provision of information and education, not only surrounding the initial implantation, but also continuously onwards, for the entirety of the implant’s lifespan inside the body, is emphasized. In this context, the communication of technological and implant-related information by healthcare professionals, as well as the appraisal skills required by patients, highlight the relevance of promoting systemic, as well as individual, technological health literacy. In order that patients do not feel disregarded and are aware of the technological innovations and advancements, designing low-threshold and accessible information offers is indicated. Here, short information channels (e.g., newsletter) to provide patients directly with new information and updates may be useful support regarding their long-term implant care and management.

A trend towards more compatibility between different technologies within and across the different implant manufacturers may reduce inequality in healthcare. This might also reduce the burden of decision making for patients by increasing their flexibility and the independence of a specific manufacturer’s productivity or willingness to innovate. More compatibility of electronic implantable devices with a greater variety of services provided by the entertainment and information industries may increase an individual’s well-being and product satisfaction. Considering that compatibility had the highest relative importance for participants in the CI-DCE, this indicates the need for further research on the everyday experiences of implant wearers regarding the practical aspects of device usage, as well as to push technological compatibility in implant development to maximize long-term patient benefits. Ongoing work on the process of implant standardization is necessary, since the implementation of standards can significantly enhance the compatibility between implants and offer various advantages for patients. In the same vein, improving implant security and longevity with reliable long-term effectivity, not only corresponds to patient preferences, but might also conserve resources by preventing reimplantation, which is favorable in terms of ecological sustainability.

### 4.2. Strengths and Limitations

The development and selection of the attributes and levels in the DCEs in this study were grounded on previous research and validated through cognitive interviews and expert exchange, ensuring their relevance, plausibility and comprehensibility. Previous comprehensive, qualitative studies of (prospective) implant wearers successfully identified and defined the relevant aspects of long-term implant care. Hence, there was a reasonable justification for using the DCE methodology for the purpose of this study. According to the current state of knowledge, this is the first attempt to elicit patient preferences regarding long-term implant care. However, some design limitations need to be considered. Firstly, for the benefit of the respondent’s efficiency, participants had to choose between two presented options, which could not depict the full complexity of the decisions within healthcare, where most of the time more options are available [15]. Secondly, in favor of anonymity, we did not assess participants’ implant status and, therefore, could not differentiate between the preferences of prospective patients from patients who already had an implant and had experienced implant care. This applies only to the GI- and CVI-DCE, since participants in the CI-DCE were recruited from the CI patient registry. However, this implies that all CI participants were cared for in the same clinic, presumably having a more similar background than CI wearers in general, impacting the generalization of these results. Thirdly, focusing on the main effects only, the effects of the level combinations were not identified. Fourthly, due to the complexity of long-term implant care, attributes other than the ones included in this study might also be relevant for a patient’s preferences regarding long-term implant care and remain neglected in the existing research. Fifthly, despite extensive recruitment efforts, only small sample sizes could be reached in the GI- and CVI-DCE impacting the statistical power of our analysis. However, it should be noted that the design of the DCEs was based on comprehensive, previous study results with the same target groups, allowing for a participatory process on the definition of the relevant topics. While being cautiously interpreted, the results from the DCEs with wearers of GI and CVI can be considered as having exploratory potential despite the small number of participants, providing impetus for further research.

## 5. Conclusions

The results from the DCEs provide valuable insights into important aspects of long-term implant care from the perspective of implant wearers, revealing possible directions for future research and implant development.

Pushing the technological compatibility of implants, implant longevity and safety are directions for future implant development in accordance with patients’ preferences. At the same time, it is important to make comprehensible and high-quality information about current developments easily available and work on improving patients’ abilities to understand such complex information. The promotion of technology-related health literacy can, therefore, be beneficial for maximizing patient benefits in long-term implant care. In addition, these findings can support the development of future implications in the context of prioritization of healthcare services or goods.

Overall, it is essential to ensure that technological advancements consider patient preferences and can be adequately transferred into the care reality of implant wearers, maximizing patient benefits. This accentuates the relevance of the participatory involvement of (prospective) implant wearers in the development processes and design of long-term implant care.

## Figures and Tables

**Table 1 ijerph-20-06358-t001:** Overview of the final attributes for the DCE in each clinical field (with attribute descriptions and level descriptions).

**COCHLEAR-DCE**				
**Attribute**	**Attribute Description**	**Level 1 (Reference)**	**Level 2**	**Level 3**
**Decision making**	Final decision on the implantation of a specific manufacturer’s CI model lies…	Only with your doctor [decisionDOC]	With you and your doctor [decisionSHARED, β_1_]	Only with you [decisionPAT, β_2_]
**Information source**	Obtaining information about developments regarding the CI and care context:	Through own research [infoRESEARCH]	Automatically by the manufacturer [infoMANU, β_3_]	Automatically by the clinic or audiologist [infoCLINIC, β_4_]
**Access to alternative treatments in the future**	Other treatment options after CI implantation, which are still being researched and may be available in the future...	Are excluded [futureoptionEXCLUDED]	Remain available [futureoptionPRESERVE, β_5_]	-
**Technological compatibility**	Compatibility with newer CI models and accessories or devices from other manufacturers	Not compatible [technUNCOMPATIBLE]	Compatible [technCOMPATIBLE, β_6_]	-
**Care relationship in aftercare**	Carrying out aftercare:	Fixed staff of professionals who are in exchange [aftercareFIXED]	Varying professionals according to specific needs [aftercareVARYING, β_7_]	-
**Education**	Education regarding adjustments, decisions and innovations regarding your CI is...	Not very comprehensive [educationLITTLE]	Very comprehensive [educationCOMPR, β_8_]	-
**GLAUCOMA-DCE**				
**Attribute**	**Attribute description**	**Level 1 (Reference)**	**Level 2**	**Level 3**
**Decision making**	Final decision on the implantation of a specific manufacturer’s CI model lies…	Only with your doctor [decisionDOC]	With you and your doctor [decisionSHARED, β_1_]	Only with you [decisionPAT, β_2_]
**Means in case of implant failure**	Corrective measure in the event that the implant does not (or no longer) work	Implant stays in the eye [failureSTAY]	Correction by means of intervention [failureCORRECTION, β_3_]	Implant removal [failureREMOVAL, β_4_]
**Probability of treatment success**	Chances of success of still not needing glaucoma medication 2 years after implantation are:	Over 50% [chancesofsuccessLOWER]	Over 75% [chancesofsuccessHIGHER, β_5_]	-
**Information source**	Information that goes beyond implantation and aftercare (e.g., on nutrition, drops for dry eyes, glasses, etc.) will be provided…	In the context of implant care from the medical side [infosourceMEDICAL]	From independent information sources (e.g., glaucoma forum) [infosourceINDEPENDENT, β_6_]	-
**Data transparency**	Statistics and empirical values on implantation in the treating clinic…	Are not available [dataNONAVAILABLE]	Are available [dataAVAILABLE, β_7_]	-
**Information exchange between health professionals**	Exchange of information between different health professionals (e.g., resident ophthalmologist, family doctor and clinic)	You coordinate yourself [infoexchangePAT]	Takes place automatically [infoexchangeAUTOM, β_8_]	-
**CARDIOVASCULAR-DCE**				
**Attribute**	**Attribute description**	**Level 1 (Reference)**	**Level 2**	**Level 3**
**Decision making**	Final decision on the implantation of a specific manufacturer’s CI model lies…	Only with your doctor [decisionDOC]	With you and your doctor [decisionSHARED, β_1_]	Only with you [decisionPAT, β_2_]
**Information source**	Information that goes beyond implantation and aftercare (e.g., on nutrition, lifestyle, etc.) will be provided…	In the context of implant care from the medical side [infoMEDICAL]	From independent information sources (e.g., German Heart Foundation) [infosourceINDEPENDENT, β_3_]	-
**Data transparency**	Statistics and empirical values on stent implantation in the treating heart clinic…	Are not available [dataNONAVAILABLE]	Are available [dataAVAILABLE, β_4_]	-
**Information exchange between health professionals**	Exchange of information between different health professionals (e.g., family doctor, clinic and other health professionals)	You coordinate yourself [infoexchangePAT]	Takes place automatically [infoexchangeAUTOM, β_5_]	-
**Means of maintaining treatment success**	The long-term success of the treatment can be secured by…	Lifelong medication (e.g., blood thinners) [successMEDICATION]	New implant every 10 years [successREIMPL, β_6_]	-
**Invasiveness of intervention**	The implant is inserted…	Minimally invasive (without opening the chest) [invasivenessMIN]	Surgical (chest opening) [invasivenessSURGERY, β_7_]	-

Note. []—Given in square brackets are the names of the variables used in the regression analysis relating to each attribute level and the corresponding beta coefficients.

**Table 2 ijerph-20-06358-t002:** Participant eligibility criteria.

Inclusion	Exclusion
(prospective) Implant wearer of respective implants	Not an (prospective) implant wearer of respective implants
Minimum age: 18 years	Age < 18 years
Consent	Does not consent to voluntary study participation or data processing

**Table 3 ijerph-20-06358-t003:** Model coefficients and odds for CI-DCE.

Attribute Levels (Ref: Reference Category)	Term in Model Equation	B	SE	Wald	df	Sig.	OR	95% Confidence Interval for OR	
								Lower	Upper
Final decision on the implantation of a specific manufacturer’s CI model lies with the patient and their doctors (ref: only doctor)	decisionSHARED	1.010	0.165	37.690	1	<0.001 **	2.745	1.989	3.789
Final decision on the implantation of a specific manufacturer’s CI model lies with the patient only (ref: only doctor)	decisionPAT	0.627	0.146	18.396	1	<0.001 **	1.872	1.406	2.494
Obtaining information about the developments regarding the CI and care context automatically by the clinic or audiologist (ref: own research)	infoCLINIC	0.569	0.110	26.576	1	<0.001 **	1.767	1.423	2.194
Obtaining information about the developments regarding the CI and care context automatically by the manufacturer (ref: own research)	infoMANU	0.340	0.128	7.084	1	0.008 *	1.404	1.094	1.803
Other treatment options after CI implantation, which are still being researched and may be available in the future remain available (ref: are excluded)	futureoptionPRESERVE	0.881	0.15	34.666	1	<0.001 **	2.414	1.800	3.237
CI is compatible with newer CI models and accessories or devices from other manufacturers (ref: no technological compatibility)	technCOMPATIBLE	1.569	0.182	74.130	1	<0.001 **	4.800	3.358	6.859
Aftercare is carried out by a variety of professionals according to specific needs (ref: fixed staff of professionals who are in exchange)	aftercareVARYING	−0.172	0.116	2.211	1	0.137	0.842	0.672	1.056
Education regarding adjustments, decisions and innovations regarding your CI is very comprehensive (ref: little comprehensive)	educationCOMPR	0.781	0.155	25.501	1	<0.001 **	2.185	1.613	2.958
	sex	−0.005	0.005	0.934	1	0.334	0.995	0.984	1.005
	age	0.000	0.000	0.023	1	0.880	1.000	1.000	1.000
	(intercept)	−2.366	0.1830	167.151	1	<0.001 **	0.094	0.066	0.134

Note. CI—cochlear implant. B—β coefficient in model Equation (1) (see Appendix B). SE—standard error. Wald—Wald chi-square statistic. df—degrees of freedom. OR—odds ratio. Ref—reference category. ** *p* < 0.001. * *p* < 0.05.

**Table 4 ijerph-20-06358-t004:** Model coefficients and odds for the GI-DCE.

Attribute Levels (Ref: Reference Category)	Term in Model Equation	B	SE	Wald	df	Sig.	OR	95% Wald Confidence Interval for OR	
								Lower	Upper
Final decision on the implantation of a specific manufacturer’s GI model lies with the patient and their doctors (ref: only doctor)	decisionSHARED	0.681	0.327	4.334	1	0.037 *	1.976	1.041	3.752
Final decision on the implantation of a specific manufacturer’s GI model lies with the patient only (ref: only doctor)	decisionPAT	0.192	0.38	0.256	1	0.613	1.212	0.576	2.549
Corrective measure in the event that the implant does not (or no longer) work: correction by means of intervention (ref: implant stays in the eye)	failureCORRECTION	0.762	0.311	6.003	1	0.014 *	2.142	1.165	3.938
Corrective measure in the event that the implant does not (or no longer) work: implant removal (ref: implant stays in the eye)	failureREMOVAL	0.250	0.273	0.843	1	0.358	1.285	0.753	2.192
Chances of success of still not needing glaucoma medication 2 years after implantation is over 75% (ref: over 50%)	chancesofsuccessHIGHER	1.714	0.440	15.167	1	<0.001 **	5.551	2.343	13,152
Information that goes beyond implantation and aftercare (e.g., on nutrition, drops for dry eyes, glasses, etc.) will be provided from independent information sources (e.g., glaucoma forum) (ref: in the context of implant care from the medical side)	infosourceINDEPENDENT	−0.814	0.315	6.684	1	0.010 *	0.443 (2.257) ^1^	0.239 (1.218) ^1^	0.821 (4.184) ^1^
Statistics and empirical values on implantation in the treating clinic are available (ref: not available)	dataAVAILABLE	0.932	0.255	13.382	1	<0.001 **	2.540	1.541	4.185
Exchange of information between different health professionals (e.g., resident ophthalmologist, family doctor and clinic): takes places automatically (ref: you coordinate yourself)	infoexchangeAUTOM	−0.053	0.227	0.056	1	0.814	0.948	0.608	1.478
	sex	−0.003	0.007	0.256	1	0.613	0.997	0.984	1.010
	age	0.001	0.000	1.373	1	0.241	1.001	1.000	1.001
	(Intercept)	−1.537	0.463	11.047	1	<0.001 **	0.215	0.087	0.532

Note. GI—glaucoma implants. B—β coefficient in model Equation (2) (see Appendix B). SE—standard error. Wald—Wald chi-square statistic. df—degrees of freedom. OR—odds ratio. Ref—reference category. ** *p* < 0.001. * *p* < 0.05. ^1^ inverted.

**Table 5 ijerph-20-06358-t005:** Model coefficients and odds for the CVI-DCE.

Attribute Levels (Ref: Reference Category)	Term in Model Equation	B	SE	Wald	df	Sig.	OR	95% Wald Confidence Interval for OR	
								Lower	Upper
Final decision on the implantation of a specific manufacturer’s CVI model lies with the patient and their doctors (ref: only doctor)	decisionSHARED	0.994	0.396	6.294	1	0.012 *	2.702	1.243	5.874
Final decision on the implantation of a specific manufacturer’s CVI model lies with the patient only (ref: only doctor)	decisionPAT	0.579	0.289	4.021	1	0.045 *	1.784	1.013	3.142
Information that goes beyond implantation and aftercare (e.g., on nutrition, lifestyle, etc.) will be provided by independent information sources (ref: in the context of implant care from the medical side)	infosourceINDEPENDENT	−0.123	0.340	0.132	1	0.716	0.884	0.454	1.721
Statistics and empirical values on implantation in the treating heart clinic are available (ref: not available)	dataAVAILABLE	0.450	0.161	7.787	1	0.005 *	1.568	1.143	2.150
Exchange of information between different practitioners (e.g., family doctor, clinic and other practitioners)	infoexchangeAUTOM	−0.057	0.248	0.053	1	0.818	0.945	0.581	1.536
The long-term success of the treatment can be secured with a new implant every 10 years (ref: lifelong medication)	successREIMPL	−2.473	0.495	24.946	1	<0.001 **	0.084 (11.9) ^1^	0.032 (4.484) ^1^	0.223 (31.25) ^1^
The implant is inserted surgically (chest opening) (ref: minimally invasive)	invasivenessSURGERY	−0.738	0.465	2.519	1	0.113	0.478	0.192	1.189
	sex	0.003	0.006	0.340	1	0.560	1.003	0.992	1.015
	age	0.000	0.000	1.886	1	0.170	1.000	1.000	1.001
	(Intercept)	0.915	0.5089	3.236	1	0.072	2.498	0.921	6.772

Note. CVI—cardiovascular implant. B—β coefficient in model Equation (3) (see Appendix B). SE—standard error. Wald—Wald chi-square statistic. df—degrees of freedom. OR—odds ratio. Ref—reference category. ** *p* < 0.001. * *p* < 0.05. ^1^ inverted.

## Data Availability

The datasets generated and analyzed during the current study, as well as the experimental design plan for each DCE and the DCE surveys are available in the open science framework (OSF) repository, https://osf.io/e2qja/ (accessed on 5 July 2023).

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
