# Peer review of "Patient Preferences for Long-Term Implant Care in Cochlear, Glaucoma and Cardiovascular Diseases"

_ijerph, 2023, doi:10.3390/ijerph20146358_

Round 1
Reviewer 1 Report
Dear Authors
Your article is of outstanding quality. You did a very good job summarizing and presenting the state-of-the-art of long-term implant care – results from discrete choice experiments in the context of cochlear, glaucoma and cardiovascular implants.
However, I believe that the conclusions should be improved with some information related to how the doctor's involvement in the patient's literacy should be approached.
Reviewer 2 Report
Review
The current study - Patient preferences for long-term implant care results from discrete choice experiments in the context of cochlear, glaucoma and cardiovascular implants
is a study that analyzes current issues, joining an upward trend in the path of scientific information.
I have a few remarks to complete this article, namely:
Title
The title is too long and is not very easy to understand. I suggest reformulation.
I suggest
The patient's perception in implant therapy in cochlear, glaucoma and cardiovascular diseases
Abstract
It must be rewritten more precisely, the purpose, material and method, how many patients, what type of treatment they received. The results are evasive and it is not clear from the conclusions what has been completed.
Introduction
The introduction is written in a general way, the prevalence of the diseases treated with implants is not presented, nor the incidence to evaluate the importance of the assessment of perception. The purpose of this work is not clearly defined. The purpose of Line 76, line 81 is repeated. It is not shown what type of study will follow. The bibliographic sources in the introduction are few in number (7) and only two source from the last five years. I suggest rewriting this section in a clear sequential manner.
Material and method
The study has batch limitations, the Alpha Coefficient (Cronbach) is not presented
which is used to study the internal consistency of a questionnaire's items. It is not shown whether the selected batch is suggestive or not. How was the sampling done?
Did the authors do a pilot study? The authors wrote that they were inspired by various studies to formulate the questions, but how did they validate the questionnaire? It does not appear if they used an already validated one.
What is the symptomatology relationship between the conditions that have been treated with various types of implants? In practice, isn't it even then to motivate the authors why they chose these types of implants? Why don't they have dental implants? However, the implants that are applied improve the quality of life of an individual or this is their purpose. What did the authors try to emphasize? That it was not a correct decision to make the implant from the point of view of the patient's perception.
Result
The results do not lead to any pertinent conclusions that would determine the change of the therapeutic plan in the future or the patient's consent for such interventions. Attention to table numbering Table 5 line 270. The data collected are from a small number of patients, they cannot be conclusive.
Discussions
In the discussion chapter, comparisons with the results of other studies are not highlighted in order to have a term of reference. They are due to the condition that was treated by the implant.
Conclusions
The conclusions are not clear, they should be reformulated.
Bibliography
There are too few sources from the last 5 years. Please improve this aspect.
Reviewer 3 Report
The authors presented a questionnaire about preferred implants and several conditions for choice. The work is quite novel, since it covers long term aspects. The article is quite well written and therefore I only recommend some minor changes.
11. Glaucoma has in Table Ref. I guess the authors wanted to add a reference
22. Utilized equations not displayed. Abbreviations and parameters in Table 1 not defined in caption.
33. There are novel coatings, which novel offer long-lived conditions and low maintenance and good possibilities to anchor them in the body. These kinds of coatings which even can adjust the attaching cell types1 and, like for cardiovascular systems2 and deep in the body might make choices in future easier, at least for the foreseeable future, at least for passive implants. The authors might mention differences between active and passive implants, as the updatability and compatibility mainly affect active implants and point coating examples out which offer future and partly already benefits compared to the past.
44. The Authors should also mention the importance of standardization for implants. These standards can significantly enhance compatibility between implants and offer advantages for patients. The authors should mention, that there is more work necessary, so if DIN / ISO or other committee members read the article they can mention it.
References
(1) Sun, Z.; Khlusov, I. A.; Evdokimov, K. E.; Konishchev, M. E.; Kuzmin, O. S.; Khaziakhmatova, O. G.; Malashchenko, V. V; Litvinova, L. S.; Rutkowski, S.; Frueh, J.; Kozelskaya, A. I.; Tverdokhlebov, S. I. Nitrogen-Doped Titanium Dioxide Films Fabricated via Magnetron Sputtering for Vascular Stent Biocompatibility Improvement. J. Colloid Interface Sci. 2022, 626, 101–112. https://doi.org/10.1016/j.jcis.2022.06.114.
(2) Hossfeld, S.; Nolte, a; Hartmann, H.; Recke, M.; Schaller, M.; Walker, T.; Kjems, J.; Schlosshauer, B.; Stoll, D.; Wendel, H.-P.; Krastev, R. Bioactive Coronary Stent Coating Based on Layer-by-Layer Technology for SiRNA Release. Acta Biomater. 2013, 9 (5), 6741–6752. https://doi.org/10.1016/j.actbio.2013.01.013.
Author Response
Please see the attachement.

Round 2
Reviewer 2 Report
Dear authors,
Thank you for submitting this article for review. Thanks to the group of authors who tried to fulfill the requirements. I believe that the article can be published in this form.
Thank you,
Regards
Dear Editor,
Thank you for submitting this article for review. Thanks to the group of authors who tried to fulfill the requirements. I believe that the article can be published in this form.
Thank you,
Regards,